# Influence of physical activity on serum vitamin D levels in people with multiple sclerosis

**Angelika Bauer[1☉], Ivan Lechner[1☉], Michael Auer[1], Thomas Berger[2], Gabriel Bsteh[2], Franziska Di Pauli[1], Harald Hegen[1], Sebastian Wurth[1], Anne Zinganell[1], Florian Deisenhammer[1]***

**1** Department of Neurology, Medical University of Innsbruck, Innsbruck, Austria, **2** Department of Neurology, Medical University of Vienna, Vienna, Austria

☉ These authors contributed equally to this work.
* florian.deisenhammer@tirol-kliniken.at

## Abstract

In most cases, multiple sclerosis (MS) patients reduce physical activity with disease progression and many patients are found to be vitamin D deficient. The aim of this study was to explore correlations between daily physical activity in everyday life and 25-hydroxyvitamin-$D_3$ ($25(OH)D_3$) serum levels in mildly disabled patients with an Expanded Disability Status Scale (EDSS) $\leq 4$. We analyzed serum $25(OH)D_3$ levels and recorded daily physical activity (activity duration, number of steps, distance, energy expenditure) using an activity tracker for 14-days in 25 women and 15 men. Participants recorded their daily sunlight exposure time by diary during the study period. We found a positive correlation between physical activity and $25(OH)D_3$ levels in both, Pearson correlation (r = 0.221) and multivariate regression analysis (β = 0.236), which was stronger than correlation with sunlight exposure time (β = -0.081). EDSS and physical activity were weakly correlated (r = -0.228), but no correlation between EDSS and $25(OH)D_3$ levels was found (r = -0.077). There were no relevant differences in physical activity (p = 0.803) and $25(OH)D_3$ concentrations (p = 0.385) between the EDSS groups 0 – 1.5 and 2.0 – 4.0. In conclusion, physical activity has an effect on vitamin D levels independent of sunlight exposure time in people with MS (pwMS) with low-grade disability.

## Introduction

Multiple sclerosis (MS) is a chronic inflammatory demyelinating disease of the central nervous system whose cause is unknown. Apart from the genetic background a large number of environmental factors have been investigated as potential risk factors for MS or disease activity in people with MS (pwMS) [1]. One of these much-discussed factors is decreased vitamin D levels which are associated with a higher risk of MS diagnosis and disease activity [2]. An inherent challenge with association and correlation is the fact that it does not prove causality and might be explained by confounders [3]. Causality can be shown by testing exposure versus non-exposure, ideally in a prospective randomized setting, which has been done for vitamin D

**Funding:** The authors received no specific funding for this work.

**Competing interests:** The authors have declared that no competing interests exist.

supplementation in pwMS in several trials, none of which could demonstrate an effect on clinical outcomes [4]. Therefore, alternative reasons for the association between MS disease activity and vitamin D levels need to be considered. As vitamin D level is driven by exposure to ultraviolet (UV) B radiation to a large extent [5] and the main source of UV-B is sunlight, one might speculate that low vitamin D levels in pwMS are the consequence of less outdoor activity due to disability.

The aim of this study was to investigate the correlation between objectively measured physical activity in everyday life as well as sunlight exposure time on 25-hydroxyvitamin-$D_3$ (25 (OH)$D_3$) serum levels in mildly disabled pwMS in a short prospective observational setting.

## Methods

For this prospective, observational study, we recruited forty pwMS at the Department of Neurology in Innsbruck. The inclusion criteria were as follows: 1) diagnosed with MS of any subtype using the diagnostic criteria that were valid at the time of diagnosis [6] with an EDSS score of $\leq 4$, defined as being fully ambulatory without the need for any walking aid [7]. 2) no vitamin D supplementation for 38 days immediately before and during the study period (due to the half-life period of approximately two to three weeks [5], it can be expected that after 38 days previous vitamin D supplementation has no effect on 25(OH)$D_3$ serum levels. 3) no vacation during the study period due to a bias in measured physical activity and time spent outdoors.

To ensure comparability of measured 25(OH)$D_3$ serum levels and sunlight exposure time, the whole study was conducted in the time between 15th August and 15th September 2017 beginning the day after blood sampling.

Using an activity tracker (AS80, Beurer Germany—BAT) on the non-dominant wrist, study participants measured their daily step-count, active time and energy expenditure (in calories) in everyday life during a period of 14 days. Steps were registered once thirty steps had been taken in a row with a maximum inactivity time of two seconds in between. Physical activity was defined by number of steps and activity time. BAT was set up individually and anonymized. Sex, age, height and current weight were registered in each device. To determine the average step length, participants had to perform a 7.62 meters' walk twice. To eliminate bias, the display of the BAT was covered up and participants were instructed not to retrieve data during the study period. BAT stored the measured values in its random-access-memory for 30 days and additionally on the manufacturer's server in Germany, fulfilling current data safety standards. The data transfer was performed after the two-week study period by using Bluetooth LE® on an Apple® iPhone with operation system iOS 10.3.1.

Further, participants had to keep a log of the time spent outdoors with sunlight exposure each day during the study period and were instructed to avoid extra efforts and unusual activities.

Levels of 25(OH)$D_3$ in serum were measured at the central laboratory of the university hospital Innsbruck using high-performance liquid chromatography (ChromSystems) and serum levels were classified according to the guidelines of the Endocrine Society, where vitamin D deficiency is defined as levels below 50 nmol/l, insufficiency as levels between 50.1–74.9 nmol/ l, sufficiency as levels between 75.0–99.9 nmol/l and ideal as levels $\geq$ 100.0 nmol/l [2].

### Statistical analyses

Statistical analyses were performed with IBM SPSS Statistics 20 (IBM Corp., Armonk, NY, USA). All data were anonymized, and the analyses of outcomes were based on the per-protocol principle. Normality of data was assessed by using the Shapiro-Wilk normality test. All

normally distributed variables are shown as mean and standard deviation (SD), while non-parametric variables are presented as median, inter-quartile range (IQR) and minimum to maximum ranges. Correlations between continuous variables were assessed by Pearson's and Spearman's correlation test as appropriate. To interpret correlation coefficients Cohen's standard classification ranges were used (weak 0.10–0.29, moderate 0.30–0.49, strong $\geq$ 0.50).

A multivariate linear regression analysis was performed to analyze the influence of physical activity and daily sun exposure time (both independent variables) on the $25(OH)D_3$ serum levels (dependent variable). Due to the small study cohort size, no further independent variables were included in the model to avoid overfitting. Cohen's $f^2$ was used as a measure of local effect size, classifying $f^2 \geq 0.02$ as small, $f^2 \geq 0.15$ as medium and $f^2 \geq 0.35$ as large effect sizes.

The $25(OH)D_3$ status and daily activity was compared between two EDSS groups (EDSS 0–1.5 vs EDSS 2.0–4.0), a two-sided t-test. The reasoning was that no impairment can be expected with EDSS scores of $\leq$ 1.5, whereas EDSS scores of $\geq$ 2.0 usually reflect physical impairment.

Non-normally distributed variables were compared between two groups with a Mann-Whitney U-test.

Due to the explorative character we did not set a level of statistical significance and consequently, no formal power calculation was performed [8,9].

## Ethics

The study was approved by the Ethics Committee of the Medical University of Innsbruck (study number 1088/2017) and all participants signed a written informed consent.

## Results

### Clinical characteristics of the participants

We included 38 patients (23 women, 15 men) and two participants were excluded. One female participant was lost to follow-up and one female participant was classified as an outlier with respect to her $25(OH)D_3$ serum level (possible laboratory error and the analysis could not be repeated). All participants had Caucasian skin type. The mean age of all 38 study participants was 39.9 years (± 9.3) and the median EDSS was 2.0. All patients had relapsing remitting disease with a mean annualized relapse rate of 0.46 including self-reported attacks (Table 1). Vitamin $D_3$ serum levels were higher in women (93.7 ± 32.1 nmol/l) compared to men (73.9 ± 19.4 nmol/l, p = 0.045). However, total energy expenditure was lower in women (women 1,680.2 ± 211.1 kcal/day vs. men 2,125.6 ± 123.9 kcal/day, p < 0.001). Sun exposure time did slightly differ between sexes with a median time of 70.4 min/day in women (range 32.5–150.0 min/day) vs. 106.9 min/day in men (range 45.0–276.4 min/day; p = 0.110). Ideal vitamin D levels were detected in 13/38 (34.2%) participants, sufficient levels in 9/38 (23.7%), insufficient levels in 14/38 (36.8%) and deficient levels in 2/38 (5.3%). Only 2 participants had to stop vitamin D supplementation before study entry whereas 38 participants never used such medication.

### Correlation between EDSS and physical activity

There was a weak negative correlation between EDSS and activity (r = -0.228) and no correlation was found between EDSS and $25(OH)D_3$ (r = -0.077). EDSS groups 0–1.5 (n = 17) and 2.0–4.0 (n = 21) did not differ regarding $25(OH)D_3$ levels (90.7 ± 36.1 nmol/l vs. 82.4 ± 21.8 nmol/l, p = 0.385) nor for daily activity time (58.4 ± 13.6 min vs. 56.6 ± 27.8 min, p = 0.803).

**Table 1. Clinical characteristics of the study participants.**

| | total | female | male |
|---|---|---|---|
| sex | | | |
| n (%) | 38 (100.0) | 23 (60.5) | 15 (39.5) |
| age (years) | | | |
| M ± SD (range) | 39.9 ± 9.3 (23.0 - 63.0) | 42.0 ± 9.7 (25.0 - 63.0) | 36.7 ± 7.9 (23.0 - 49.0) |
| clinical phenotype | | | |
| n (%) | RRMS: 38 (100.0) | | |
| EDSS | | | |
| median (IQR, range) | 2.0 (1.5, 0 - 4) | 2.0 (1.5, 0 - 4) | 1.5 (1.0, 0 - 3) |
| disease duration (months) | | | |
| M ± SD (range) | 131.4 ± 96.3 (9 - 352) | 175.6 ± 92.9 (9 - 352) | 63.5 ± 52.8 (11 - 168) |
| annualized relapse rate* | | | |
| M ± SD (range) | 0.46 ± 0.48 (0.00–2.67) | 0.41 ± 0.57 (0.00–2.67) | 0.53 ± 0.30 (0.00–1.09) |
| 25(OH)D$_3$ (nmol/l) | | | |
| M ± SD (range) | 86.1 ± 28.9 (44.0 - 173.0) | 93.7 ± 32.1 (44.0 - 173.0) | 73.9 ± 19.4 (52.0 - 104.0) |
| activity/day (min) | | | |
| M ± SD (range) | 57.4 ± 22.3 (16.0 - 120.0) | 57.0 ± 23.4 (16.0 - 120.0) | 59.1 ± 21.6 (26.0 - 113.0) |
| steps/day | | | |
| M ± SD (range) | 6,863.3 ± 2,592.0 (2,082.9 - 14,656.0) | 6,862.9 ± 2,785.9 (2,082.9 - 14,656.0) | 7,003.8 ± 2,381.5 (3,293.4 - 13,060.7) |
| distance/day (km) | | | |
| median (IQR, range) | 4.2 (3.0, 1.3 - 9.9) | 4.0 (2.9, 1.3 - 9.7) | 5.5 (2.9, 2.6 - 9.9) |
| total calorie consumption/day (kcal) | | | |
| M ± SD (range) | 1,852.1 ± 281.0 (1,292.1 - 2,373.8) | 1,680.2 ± 211.1 (1,292.1 - 2,083.9) | 2,125.6 ± 123.9 (1,944.0 - 2,373.8) |
| sun exposure time/day (min) | | | |
| median (IQR, range) | 72.9 (71.8, 32.5 - 276.4) | 70.4 (60.3, 32.5 - 150.0) | 106.9 (65.1, 45.0 - 276.4) |
| fish consumption/week (g) | | | |
| median (IQR, range) | 150.0 (150.0, 0 - 700.0) | 75.0 (150.0, 0 - 400.0) | 200 (106.0, 0 - 700.0) |
| BMI (kg/m$^2$) | | | |
| M ± SD (range) | 23.8 ± 4.2 (16.5 - 34.5) | 23.5 ± 4.8 (16.5 - 34.5) | 24.3 ± 3.3 (18.4 - 31.1) |

Abbreviations: IQR, interquartile range; M ± SD, mean ± standard deviation; n, number; EDSS, expanded disability status scale; 25(OH)D3, 25-hydroxyvitamin-D3; BMI, Body mass index

* up to 5 years before study entry depending on disease duration, includes self-reported relapses.

## Correlation between physical activity and 25(OH)D$_3$ levels

A weak correlation between activity and 25(OH)D$_3$ levels was found (r = 0.221; Fig 1 panel A). Further, a weak negative correlation was found between body mass index (BMI) and 25(OH)D$_3$ (r = -0.282) and a moderate negative correlation between body weight and 25(OH)D$_3$ (r = -0.403).

Sun exposure time and 25(OH)D$_3$ levels were not correlated (r = -0.002), possibly due to a ceiling effect as discussed below (Fig 1 panel B). However, many other factors might play a role such as skin type, genetic make-up, clothing, use of sun blockers, age, and altitude. Performed steps and daily activity were strongly correlated (r = 0.989), and there was a moderate correlation between walked distance and total energy expenditure (r = 0.446).

## Regression analyses

The influence of physical activity and daily sun exposure time on 25(OH)D$_3$ serum levels was analyzed with a multivariate linear regression model: Combined, both variables (physical

Panel A

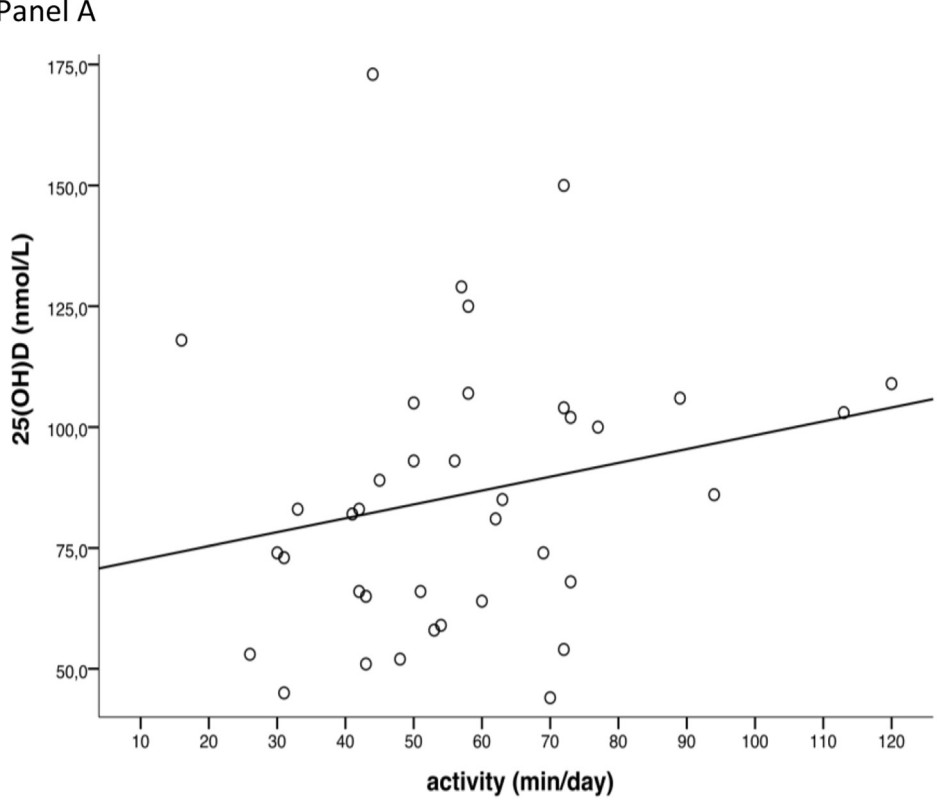

Panel B

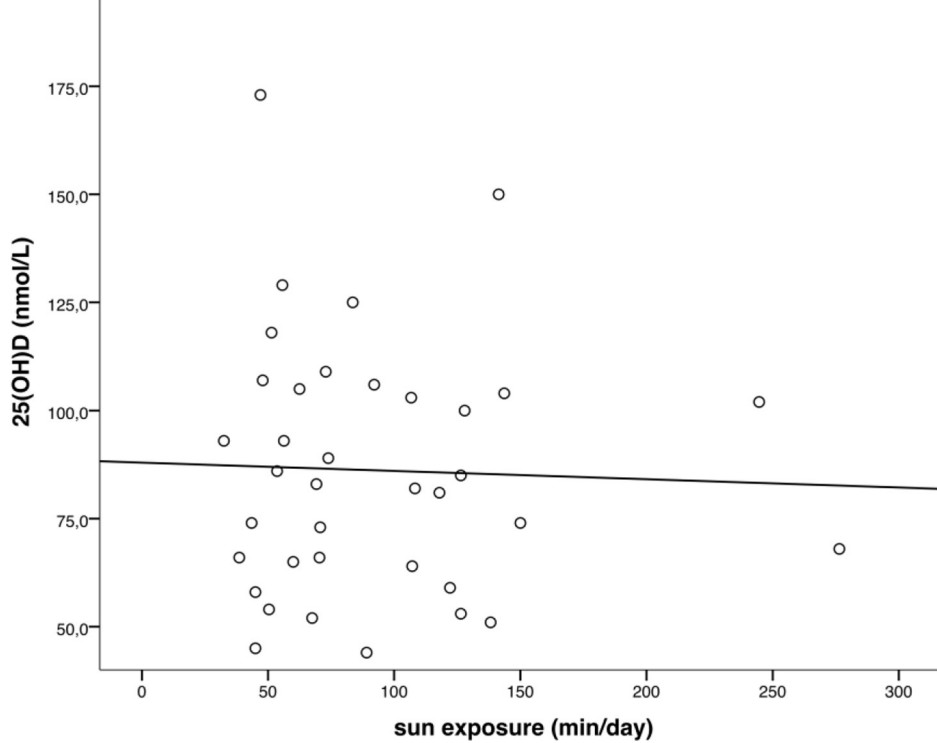

**Fig 1.** Scatter plots and linear regression lines for vitamin D concentrations versus activity (r = 0.221; panel A) and sun exposure time (r = -0.002; panel B).

activity and sun exposure time) had a weak influence on $25(OH)D_3$ levels ($R^2$ = 0.055, p = 0.383). Furthermore, local effect size was classified as being small, using Cohen's $f^2$ ($f^2$ = 0.058).

Importantly, the standardized coefficient (β) indicated that physical activity has a stronger influence on $25(OH)D_3$ levels than daily sun exposure time (β = 0.236, p = 0.174 vs. β = -0.081, p = 0.638, Table 2). No interference between physical activity and sun exposure time in multivariate linear regression analyses could be detected.

## Discussion

We investigated the effect of objectively measured physical activity on $25(OH)D_3$ serum levels in pwMS who had EDSS scores ≤ 4 and observed a correlation between daily activity time and $25(OH)D_3$ serum levels independent of sun exposure time. This is in line with a previous study, also finding a trend towards higher $25(OH)D_3$ serum levels in more active pwMS with an EDSS ≤ 3.5 evaluating the activity status by questionnaire and measurement of cardiorespiratory fitness using spiroergometry, an objective measure of physical performance [10]. The authors assumed that higher vitamin D levels in active people were driven through longer sunlight exposure [10] because the majority of the study population reported longer outdoor rather than indoor activity. In contrast, we found no association between sunlight exposure time and vitamin D serum levels. This may be explained by a ceiling effect of serum $25(OH)D_3$ levels after cumulative UVB radiation time in summer but not by previous vitamin D supplementation as the vast majority did not take any such medication. The steady state between vitamin D production and degradation is already reached within 20 min of UV-radiation exposure in Caucasians and with further UVB exposure previtamin $D_3$ is photoisomerized to the biologically inactive isomers lumisterol$_3$ and tachysterol$_3$ [11,12]. The ceiling concentration of serum $25(OH)D_3$ was obtained at levels of 55–80 nmol/l through phototherapy [12] comparable to our study participants with an average level of 86 nmol/l. The slight difference may be explained by use of different assays [13]. There is a number of other possible confounders that may influence vitamin D concentrations, such as skin type, genetic make-up, clothing, use of sunscreen, age and altitude [14,15]. It is hard to determine how much these factors contributed to the results which were not included or not recorded to avoid overfitting of regression analyses limiting interpretation of our observations. Similar to our findings others observed a correlation between activity and plasma $25(OH)D_3$ concentrations independent of sun exposure time or outdoor exercise [14,16] indicating that physical activity can raise vitamin D levels independent of sunlight exposure. How vitamin D levels are driven by physical activity remains a matter of debate. There is some evidence that parathyroid hormone, which is stimulated by exercise, activates renal calcitriol synthesis [17]. Also, the exercise-induced

**Table 2. Multivariate and univariate linear regression analyses.**

| Multivariate linear regression (influence of sun exposure and physical activity on $25(OH)D_3$ levels) | Univariate linear regression (influence of sun exposure or physical activity on $25(OH)D_3$ levels) |
|---|---|
| $R^2$ = 0.055, p = 0.383 | |
| Sun exposure: β = -0.081, p = 0.638 | Sun exposure: β = -0.035, p = 0.837 |
| Physical activity: β = 0.236, p = 0.174 | Physical activity: β = 0.221, p = 0.183 |

decrease of serum phosphate (a vitamin D inhibitor) might lead to an increase of vitamin D levels [18].

In some studies, higher vitamin D levels were attributed to activity during sunlight exposure, however, in most of these investigations physical activity independent of sunlight exposure was not accounted for. In the present study, pwMS documented 1.2 hours of sunlight exposure per day on average, which is similar to another study reporting 1.4 hours, also using a diary [19].

Our findings of high BMI and higher body weight being negatively correlated with vitamin D levels are consistent with other studies [20,21]. This can be partially explained by the dilution of intracutaneously synthesized or ingested vitamin D in adipose tissue [22] and by less UVB radiation exposure because of less participation in outdoor activities [23]. Furthermore, solid evidence suggests that high BMI levels, particularly high levels of adipose tissue induce inflammation [23]. This ongoing inflammation processes in pwMS may consume vitamin D and hence, support our findings [23,24].

Past studies addressing the daily number of steps in pwMS are inconsistent with conflicting results even in patients on a similar level of disability, presumably because different tracking devices were used. This led to a wide range of documented average step counts, starting from 5,903 ± 3,185 up to 10,243 ± 3,817 steps per day among fully ambulatory pwMS [25,26]. In our study cohort we observed an average step count (6,863 ± 2,592) at the lower end of the range described above. Likewise, the total energy expenditure in our study cohort was comparably low. The BAT device probably underestimates true physical activity as it starts counting steps only after a minimum of thirty subsequent steps allowing a maximum interval of two seconds between each step. Despite these drawbacks we selected BAT over other tracking devices due to the internal memory and battery capacities. However, because all participants used the same device comparability within our study cohort is given. We are aware of further limitations, particularly the relatively small sample size because of the explorative purpose of the study and the weak to moderate effect sizes. However, our results might be considered in future interventional vitamin D trials, not only in MS, by including activity as a covariate.

Although the sample size was low due to the explorative setting, the present study indicates that physical activity correlates with 25(OH)D$_3$ serum levels in ambulatory pwMS. Previous studies could demonstrate a positive impact of physical activity on various clinical outcomes in pwMS [27,28]. Apart from that, physical activity might have an effect on vitamin D serum levels, whereas vitamin D supplementation can increase 25(OH)D$_3$ serum levels but failed to show an effect on primary clinical outcomes in a series of interventional trials [4,29,30].

## Supporting information

**S1 Data.**
(XLSX)

## Acknowledgments

The authors would like to thank all participating pwMS for their cooperation; We thank A. Neuner and her team in the Innsbruck MS outpatient clinic for their help in collecting blood samples.

## Author Contributions

**Conceptualization:** Angelika Bauer, Florian Deisenhammer.

**Formal analysis:** Angelika Bauer, Ivan Lechner.

**Investigation:** Angelika Bauer, Ivan Lechner, Michael Auer, Franziska Di Pauli, Harald Hegen, Anne Zinganell, Florian Deisenhammer.

**Methodology:** Florian Deisenhammer.

**Writing – original draft:** Angelika Bauer, Ivan Lechner, Florian Deisenhammer.

**Writing – review & editing:** Michael Auer, Thomas Berger, Gabriel Bsteh, Franziska Di Pauli, Harald Hegen, Sebastian Wurth, Anne Zinganell.

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
