## [Decision Letter · Decision Letter 0]

21 Apr 2020

PONE-D-20-10627

Influence of physical activity on serum vitamin D levels in people with multiple sclerosis

PLOS ONE

Dear Prof. Deisenhammer,

Thank you for submitting your manuscript to PLOS ONE. After careful consideration, we feel that it has merit but does not fully meet PLOS ONE’s publication criteria as it currently stands. Therefore, we invite you to submit a revised version of the manuscript that addresses the points raised during the review process.

Please take note of the comments for the reviewer. For this to be considered there needs an extensive acknowledgement of the limitations of the study.

We would appreciate receiving your revised manuscript by Jun 05 2020 11:59PM. To enhance the reproducibility of your results, we recommend that if applicable you deposit your laboratory protocols in protocols.io, where a protocol can be assigned its own identifier (DOI) such that it can be cited independently in the future. For instructions see: http://journals.plos.org/plosone/s/submission-guidelines#loc-laboratory-protocols

We look forward to receiving your revised manuscript.

Kind regards,

Sreeram V. Ramagopalan

Academic Editor

PLOS ONE

Reviewers' comments:

Reviewer's Responses to Questions

**Comments to the Author**

1. Is the manuscript technically sound, and do the data support the conclusions?

Reviewer #1: No

2. Has the statistical analysis been performed appropriately and rigorously? 

Reviewer #1: No

3. Have the authors made all data underlying the findings in their manuscript fully available?

Reviewer #1: No

4. Is the manuscript presented in an intelligible fashion and written in standard English?

Reviewer #1: Yes

5. Review Comments to the Author

Reviewer #1: Main summary

The authors conducted a 14-days cohort-study among N=38 people with MS, with an EDSS score<=4. The aim was to assess correlations between 25(OH)D an both physical activity as well as sunlight exposure. Physical activity was measured with an electronic activity tracker, outdoor sunlight exposure with a diary. After bloodsampling, patients were followed for 14 days.

The authors report a weak negative correlation between EDSS score and acitivity (R=-0.228) but not 25(OH)D (R=-0.077).

A weak correlation between 25(OH)D levels and activity (R=0.221) but not sun exposure (R=-0.002). In a linear regression model, sunlight exposure showed in the univariate and multivariate model a less steep beta when compared to physical activity.

The authors conclude they observed no association between sunlight exposure time and serum 25(OH)D levels. They conclude that physical activity has an independent effect on MS. They conclude their paper stating that patients should be encourages to practice physical exercise rather than taking vitamin D supplements.

General comments

The authors address a relevant question, assessing the association between MS disease outcomes and vitamin D levels. However, their study is flawed by a small sample size, methodological shortcomings ignoring this small sample size, inaccurate measurements of endpoints, and an overinterpretation of own data based on earlies studies.

Major points

The authors refer to a very provocative paper of McShane et al. (Am stat 2019), justifying their decision not to set a significance level. There are several points to make regarding this statement:

1. McShane et al. argue that, in biological sciences, effects observed are usually small and variable, and that therefore a significance level of 0.05 does not per se reflect the validity of an association. The point they want to make therefore concerns mostly effect size and not sample size. In their paper, the authors reason that authors should decide whether their design (including sample size) would be accurate enough to detect an effect they want to measure. Leaving a P-value threshold is proposed as a cure for detecting small yet biologically relevant associations among small but biologically irrelevant findings, but not as a cure for underpowered studies. The decision of the authors to report in an underpowered sample correlation coefficients without adhering to any significance level is therefore not supported by the cited paper.

2. The paper by McShane et al reasons that any form of threshold-setting is not favorable (including Cohen’s classification of correlation ranges), but rather advocates that reviewers assess whether study design and data quality are robust enough to substantiate conclusions drawn including the association with prior evidence. The authors do not show scatterplots which allow the readers to assess the distribution of datapoints (including the proposed linearity of correlations), and to decide whether these datapoints convincingly support the claim of (absence of) an effect the authors propose.

3. The authors state that no power-analysis for the assumptions to be tested could be provided since there were no prior data, yet refer to 2 earlier papers in which a correlation between activity and plasma 25(OH)D concentrations independent of time spend outside was performed (Touvier et al., J Invest Dermatol 2015; Brock et al., JSBMB 2010). These data could have been used to provide estimates.

The authors refer in a variable terminology to the sunlight exposure-correlate they measured. The most accurate is probably ‘diary of time spent outdoors’ but also sunlight exposure and UVB exposure have been mentioned. The inducer of vitamin D3 photosynthesis in the skin is UVB light. Sophisticated UVB light measuring devices have been used in previous studies in MS (work of the van der Mei group, Tasmania). Besides time spent outside and BMI, UVB exposure is determined by skin type, genetic constitution, clothing, wearing sunblock, age, and altitude. In their sample of N=38, there are too many confounders to be able to conclude that no association between sunlight exposure and 25(OH)D levels is present.

The advice in the final paragraph of the paper is not in any way supported by the data which the authors collected. None of the clinical trials performed thus far has been designed to detect any effect on EDSS-score or walking ability as primary endpoint, but rather inflammatory disease activity (such as relapses or MRI-activity).

Minor points

Since the study was conducted in 2 different cities, differences between including sites should preferably be excluded (more or less sunny during sampling period, local UVB-levels are available from public sources).

The authors included a heterogeneous cohort, regarding age. The authors should also disclose characteristics as disease duration, clinical phenotype, and relapse/ MRI disease activity to provide a feeling of the patient-type included.

6. PLOS authors have the option to publish the peer review history of their article (what does this mean?). If published, this will include your full peer review and any attached files.

Reviewer #1: No

---

## [Author Response · Author response to Decision Letter 0]

6 May 2020

Response to reviewer #1

Reviewer #1: Main summary

The authors conducted a 14-days cohort-study among N=38 people with MS, with an EDSS score<=4. The aim was to assess correlations between 25(OH)D an both physical activity as well as sunlight exposure. Physical activity was measured with an electronic activity tracker, outdoor sunlight exposure with a diary. After bloodsampling, patients were followed for 14 days.

The authors report a weak negative correlation between EDSS score and acitivity (R=-0.228) but not 25(OH)D (R=-0.077).

A weak correlation between 25(OH)D levels and activity (R=0.221) but not sun exposure (R=-0.002). In a linear regression model, sunlight exposure showed in the univariate and multivariate model a less steep beta when compared to physical activity.

The authors conclude they observed no association between sunlight exposure time and serum 25(OH)D levels. They conclude that physical activity has an independent effect on MS. They conclude their paper stating that patients should be encourages to practice physical exercise rather than taking vitamin D supplements.

Response: Thanks for the concise summary.

General comments

The authors address a relevant question, assessing the association between MS disease outcomes and vitamin D levels. However, their study is flawed by a small sample size, methodological shortcomings ignoring this small sample size, inaccurate measurements of endpoints, and an overinterpretation of own data based on earlies studies.

Response: Points well taken. We address all these issues below.

Major points

The authors refer to a very provocative paper of McShane et al. (Am stat 2019), justifying their decision not to set a significance level. There are several points to make regarding this statement:

Response: We agree that there is a controversial discussion about using statistical significance. This reference has been used in lieu of a long list of other possible references addressing the same issue, but for space saving reasons decided against it. However, we have added another reference supporting the thought of not setting a level of significance because of frequent misinterpretation, particularly with respect to true versus no effect. Also, we don’t feel that paper by McShane et al is too provocative. The debate dates back several decades (e.g. Rozenboom WW, The Fallacy of the Null Hypothesis Significance Test, Psychological Bulletin, 1960; 57, 416–428) and there is a long list of scientific journals discouraging authors to use significance levels and even reporting p-values. In the instructions of GraphPad’s Prism, a very popular statistical program, it says “The entire construct of 'hypothesis testing' leading to a conclusion that a result is or is not 'statistically significant' makes sense in situations where you must make a firm decision based on the results of one P value. While this situation occurs in quality control and maybe with clinical trials, it rarely occurs with basic research.” (https://www.graphpad.com/guides/prism/6/statistics/statistical_significance_in_science.htm). Accordingly, we don’t see an urgent need to come to a firm and final decision. We rather see it as another piece of evidence supporting the positive effects of physical activity in MS.

1. McShane et al. argue that, in biological sciences, effects observed are usually small and variable, and that therefore a significance level of 0.05 does not per se reflect the validity of an association. The point they want to make therefore concerns mostly effect size and not sample size. In their paper, the authors reason that authors should decide whether their design (including sample size) would be accurate enough to detect an effect they want to measure. Leaving a P-value threshold is proposed as a cure for detecting small yet biologically relevant associations among small but biologically irrelevant findings, but not as a cure for underpowered studies. The decision of the authors to report in an underpowered sample correlation coefficients without adhering to any significance level is therefore not supported by the cited paper.

Response: We agree with the reviewer. It was not our intention to avoid a level of significance BECAUSE of lack of power. As the alpha error is one of the input parameters for a formal power calculation it could not be done. The wording in the statistical methods section was indeed unfortunate and we now made clear statements in this regard. 

2. The paper by McShane et al reasons that any form of threshold-setting is not favorable (including Cohen’s classification of correlation ranges), but rather advocates that reviewers assess whether study design and data quality are robust enough to substantiate conclusions drawn including the association with prior evidence. The authors do not show scatterplots which allow the readers to assess the distribution of datapoints (including the proposed linearity of correlations), and to decide whether these datapoints convincingly support the claim of (absence of) an effect the authors propose.

Response: Good point. We have added scatterplots for better visualisation of the data.

3. The authors state that no power-analysis for the assumptions to be tested could be provided since there were no prior data, yet refer to 2 earlier papers in which a correlation between activity and plasma 25(OH)D concentrations independent of time spend outside was performed (Touvier et al., J Invest Dermatol 2015; Brock et al., JSBMB 2010). These data could have been used to provide estimates.

Response: We agree that it looks like an apparent discrepancy. As pointed out above, we have now changed the wording and reasoning regarding power. 

4. The authors refer in a variable terminology to the sunlight exposure-correlate they measured. The most accurate is probably ‘diary of time spent outdoors’ but also sunlight exposure and UVB exposure have been mentioned. The inducer of vitamin D3 photosynthesis in the skin is UVB light. Sophisticated UVB light measuring devices have been used in previous studies in MS (work of the van der Mei group, Tasmania). Besides time spent outside and BMI, UVB exposure is determined by skin type, genetic constitution, clothing, wearing sunblock, age, and altitude. In their sample of N=38, there are too many confounders to be able to conclude that no association between sunlight exposure and 25(OH)D levels is present.

Response: Thanks for these important points. We have changed the wording regarding outdoor time uniformly throughout the manuscript to sun exposure time. There is absolutely no doubt that UVB light is the inducer of vitamin D3. It is just that we couldn’t observe a correlation between sunlight exposure time and vitamin D levels in the setting of the study, very likely because of the seasonal ceiling effect not allowing a further increase. We have addressed this issue in the discussion and added a corresponding statement in the results section. We are aware of the many possible confounders but did not include those in the multivariate model in order to avoid overfitting.

5. The advice in the final paragraph of the paper is not in any way supported by the data which the authors collected. None of the clinical trials performed thus far has been designed to detect any effect on EDSS-score or walking ability as primary endpoint, but rather inflammatory disease activity (such as relapses or MRI-activity).

Response: We agree that the advice in the last paragraph is not fully supported by our own data. We changed the wording accordingly and removed the heading “conclusion” to allow a broader discussion at the end of the manuscript.

True, the interventional clinical trials examining the effects of vitamin D in MS patients used various primary endpoints with relapse rates most frequently reported. Relapse rates are still the primary target in RCTs for relapsing MS and therefore also valid as primary outcome in vitamin D studies involving relapsing MS patients. However, there were a few studies using EDSS or MRI as primary endpoints, as reviewed in: Cochrane Database Syst Rev. 2018 Sep; 2018(9): CD008422. As outlined in the manuscript, none of these studies met the primary endpoint which is why we made a strong statement in favour of physical activity.

Minor points

Since the study was conducted in 2 different cities, differences between including sites should preferably be excluded (more or less sunny during sampling period, local UVB-levels are available from public sources).

Response: This study was not done in 2 different cities. Maybe the impression came up because two of the co-authors moved from Innsbruck to Vienna and are now using their current affiliation. The study was exclusively performed at the Medical University of Innsbruck.

The authors included a heterogeneous cohort, regarding age. The authors should also disclose characteristics as disease duration, clinical phenotype, and relapse/ MRI disease activity to provide a feeling of the patient-type included.

Response: We added this information to table 1 and the results section except for MRI activity. The latter was not part of the study protocol and we only perform MRI on particular request (e.g. unexpected events, change or end of treatment, etc) which is why we don’t have systemic data in our population.

Also, we discovered a minor mistake of the calculation of distance/day not affecting the overall results and conclusions, and changed the data in the table accordingly.

---

## [Decision Letter · Decision Letter 1]

18 May 2020

PONE-D-20-10627R1

Influence of physical activity on serum vitamin D levels in people with multiple sclerosis

PLOS ONE

Dear Prof. Deisenhammer,

Thank you for submitting your manuscript to PLOS ONE. After careful consideration, we feel that it has merit but does not fully meet PLOS ONE’s publication criteria as it currently stands. Therefore, we invite you to submit a revised version of the manuscript that addresses the points raised during the review process.

ACADEMIC EDITOR: 

In order to accept the manuscript the limitations need to be more carefully spelt out. This study is explorative, not hypothesis testing and no conclusions can be made.

We would appreciate receiving your revised manuscript by Jul 02 2020 11:59PM. To enhance the reproducibility of your results, we recommend that if applicable you deposit your laboratory protocols in protocols.io, where a protocol can be assigned its own identifier (DOI) such that it can be cited independently in the future. For instructions see: http://journals.plos.org/plosone/s/submission-guidelines#loc-laboratory-protocols

We look forward to receiving your revised manuscript.

Kind regards,

Sreeram V. Ramagopalan

Academic Editor

PLOS ONE

Reviewers' comments:

Reviewer's Responses to Questions

**Comments to the Author**

1. If the authors have adequately addressed your comments raised in a previous round of review and you feel that this manuscript is now acceptable for publication, you may indicate that here to bypass the “Comments to the Author” section, enter your conflict of interest statement in the “Confidential to Editor” section, and submit your "Accept" recommendation.

Reviewer #1: (No Response)

2. Is the manuscript technically sound, and do the data support the conclusions?

Reviewer #1: Partly

3. Has the statistical analysis been performed appropriately and rigorously? 

Reviewer #1: No

4. Have the authors made all data underlying the findings in their manuscript fully available?

Reviewer #1: Yes

5. Is the manuscript presented in an intelligible fashion and written in standard English?

Reviewer #1: Yes

6. Review Comments to the Author

Reviewer #1: The authors submitted a revision of their manuscript, in which some minor textual changes were made, a figure was added, and the table was updated. The addition of the scatterplots is very helpful, the addition of clinical data is also very helpful to disclose that their cohort is not only very small but also very heterogeneous in many aspects.

Three major points remain:

1. The issue of not setting a significance level has not been solved. The authors correctly refer to other publications raising the discussion on the use of P-values, and reveal the instructions of a popular statistical program apparently provide some guidance. The severe lack of power of their dataset still receives a very limited role in the discussion of the validity of their results. The authors should at least be very careful in their formulation of study design and of conclusions they can and cannot draw from their data. This is most critical in their abstract:

*”The aim […] was to correlate” is less accurate than “The aim […] to explore correlations between”

*”In conclusion, physical activity has an independent effect on vitamin D levels” is less accurate than “in conclusion, physical activity has an effect on vitamin D levels independent of sunlight exposure time” since many other confounders were not assessed.

2. In their explanation of a lack of correlation between 25(OH)D and sun exposure time in the manuscript, the authors completely ignore that they have neglected a lot of potential very relevant confounders (see my first comments) and their proxy estimate of UVB light exposure may be very inaccurate. Explaining the lack of correlation only as ceiling effect is not valid. The authors should include a statement on this both at line 146 (were hey attribute the lack of any correlation to ceiling effect) and in the discussion section.

3. I still have problems with the revised last line of their discussion: “Therefore, there is a dual effect of physical exercise on both, vitamin D serum levels and clinical improvement, whereas vitamin D supplementation can increase 25(OH)D3 serum levels but failed to show a clinical effect in a series of interventional trials.” This statement is still not supported by their data, further downtuning would be appropriate.

First, the authors did not show there is an effect of exercise on 25(OH)D levels in their cohort, but rather that there is a weak (not talking about significance) correlation between physical activity and 25(OH)D levels in their small cohort.

Second, the authors refer to a meta-analysis from 2018 on vitamin D supplementation studies, but do ignore the two largest and most recently published trials on vitamin D supplementation in MS (Camu et al., N3 2018; Hupperts et al., Neurology 2018). The authors are correct on the observation that these studies did not make their primary endpoints. However in secondary endpoints, and subgroup analysis, potential relevant signals were noted in relapse rate and MRI outcomes. The notion on MRI endpoints is further in line with an earlier small study by Soilu-Hanninen et al., JNNP 2012.

7. PLOS authors have the option to publish the peer review history of their article (what does this mean?). If published, this will include your full peer review and any attached files.

Reviewer #1: No

---

## [Author Response · Author response to Decision Letter 1]

22 May 2020

Response to reviewer:

The authors submitted a revision of their manuscript, in which some minor textual changes were made, a figure was added, and the table was updated. The addition of the scatterplots is very helpful, the addition of clinical data is also very helpful to disclose that their cohort is not only very small but also very heterogeneous in many aspects.

Response: Thanks for carefully evaluating the revised manuscript.

1. The issue of not setting a significance level has not been solved. The authors correctly refer to other publications raising the discussion on the use of P-values, and reveal the instructions of a popular statistical program apparently provide some guidance. The severe lack of power of their dataset still receives a very limited role in the discussion of the validity of their results. The authors should at least be very careful in their formulation of study design and of conclusions they can and cannot draw from their data. This is most critical in their abstract:

*”The aim […] was to correlate” is less accurate than “The aim […] to explore correlations between”

*”In conclusion, physical activity has an independent effect on vitamin D levels” is less accurate than “in conclusion, physical activity has an effect on vitamin D levels independent of sunlight exposure time” since many other confounders were not assessed.

Response: We agree that the wording should be more careful and adopted the suggestions accordingly. 

2. In their explanation of a lack of correlation between 25(OH)D and sun exposure time in the manuscript, the authors completely ignore that they have neglected a lot of potential very relevant confounders (see my first comments) and their proxy estimate of UVB light exposure may be very inaccurate. Explaining the lack of correlation only as ceiling effect is not valid. The authors should include a statement on this both at line 146 (were hey attribute the lack of any correlation to ceiling effect) and in the discussion section.

Response: We acknowledge the issue of confounders and have now added statements in the results and discussion sections. It is not that we ignored that fact, but it is hard to estimate how much of an effect those factors had.

3. I still have problems with the revised last line of their discussion: “Therefore, there is a dual effect of physical exercise on both, vitamin D serum levels and clinical improvement, whereas vitamin D supplementation can increase 25(OH)D3 serum levels but failed to show a clinical effect in a series of interventional trials.” This statement is still not supported by their data, further downtuning would be appropriate.

First, the authors did not show there is an effect of exercise on 25(OH)D levels in their cohort, but rather that there is a weak (not talking about significance) correlation between physical activity and 25(OH)D levels in their small cohort.

Response: True, we did not show an effect in terms of an interventional trial. The term effect is frequently used in context with effect size, strength of correlation or association, etc. However, we have changed the wording to correlation rather than effect.

Second, the authors refer to a meta-analysis from 2018 on vitamin D supplementation studies, but do ignore the two largest and most recently published trials on vitamin D supplementation in MS (Camu et al., N3 2018; Hupperts et al., Neurology 2018). The authors are correct on the observation that these studies did not make their primary endpoints. However, in secondary endpoints, and subgroup analysis, potential relevant signals were noted in relapse rate and MRI outcomes. The notion on MRI endpoints is further in line with an earlier small study by Soilu-Hanninen et al., JNNP 2012.

Response: We have added the 2 recently published studies by Camu et al and Hupperts et al. and changed the wording to “no effect on primary clinical outcomes”. As the reviewer rightly observes, the primary endpoints were not met in any of the interventional trials so far. We feel that evidence from primary outcomes outweigh secondary non-clinical outcomes and post-hoc subgroup analyses. A 5% significance level, which is most commonly used, by definition results in one out of 20 false positive results. The more post-hoc analyses one performs the higher the likelihood of such a false positive outcome. Altogether, 14 interventional trials (12 summarized in the Cochrane review, reference 29, and the 2 recent trials mentioned above) failed to meet the primary outcomes.

---

## [Editor Report · Decision Letter 2]

26 May 2020

Influence of physical activity on serum vitamin D levels in people with multiple sclerosis

PONE-D-20-10627R2

Dear Dr. Deisenhammer,

We are pleased to inform you that your manuscript has been judged scientifically suitable for publication and will be formally accepted for publication once it complies with all outstanding technical requirements.

With kind regards,

Sreeram V. Ramagopalan

Academic Editor

PLOS ONE
---

## [Editor Report · Acceptance letter]

1 Jun 2020

PONE-D-20-10627R2 

Influence of physical activity on serum vitamin D levels in people with multiple sclerosis 

Dear Dr. Deisenhammer:

I am pleased to inform you that your manuscript has been deemed suitable for publication in PLOS ONE. Congratulations! Your manuscript is now with our production department. 

With kind regards,

on behalf of

Dr. Sreeram V. Ramagopalan 

Academic Editor

PLOS ONE